# Evaluation of Corpus Luteum and Plasma Progesterone the Day before Embryo Transfer as an Index for Recipient Selection in Dairy Cows

**DOI:** 10.3390/vetsci10040262

**Published:** 2023-03-29

**Authors:** Woojae Choi, Younghye Ro, Eunhui Choe, Leegon Hong, Wonyou Lee, Danil Kim

**Affiliations:** 1Department of Farm Animal Medicine, College of Veterinary Medicine, Seoul National University, Seoul 08826, Republic of Korea; 2Farm Animal Clinical Training and Research Center, Institutes of Green-Bio Science and Technology, Seoul National University, Pyeongchang 25354, Gangwon-do, Republic of Korea; 3Lartbio Co., Ltd., Gangnam-gu, Seoul 06221, Republic of Korea

**Keywords:** embryo transfer, pregnancy rate, progesterone, corpus luteum, estrus synchronization

## Abstract

**Simple Summary:**

The use of embryo transfer (ET) in dairy cows helps to improve herd genetics. In order to improve pregnancy rates after ET, the selection of a recipient as well as a high-graded embryo is important. A larger CL and higher P4 on the day of ET represent high reproductive performance in cows. In this study, we evaluated CL size and P4 level the day before ET and their effects on the pregnancy rate. The recipients’ parity and the seasons in which ET was performed had a significant effect on the pregnancy rate, but the synchronization methods and the pre-ET evaluation values of CL size and P4 level could not predict the pregnancy rate. Embryo implantation and pregnancy maintenance depend on high P4 concentration according to the function of the CL, but in the mid-luteal phase, in which ET is performed, development variation makes it difficult to evaluate those values. Intensive evaluation of the variables involved in ET in dairy cows will lead to an improvement in the pregnancy rate and productivity.

**Abstract:**

The aim of the present study was to evaluate the effects of synchronization method, season, parity, corpus luteum (CL) size, and progesterone (P4) levels on the pregnancy rate after bovine embryo transfer (ET). Among 165 recipient candidates who received 1 of 2s estrus synchronization treatments, 96 heifers and 43 cows were selected through rectal examination and used as recipients. The day before ET, the CL size and plasma P4 concentration were evaluated. The CL sizes and plasma P4 levels were not different between the selected and unselected candidates, and the pregnancy rates with the two synchronization methods were not different. However, the pregnancy rates were higher in heifers than in lactating cows, and also higher after ET performed from September to February than from March to August (*p* < 0.05). The recipients with a CL larger than 1.5 cm showed statistically higher pregnancy rates, and although there was no statistical significance, the pregnancy rate was higher when the plasma P4 levels were between 2.0 and 4.0 ng/mL. Exposure to a stressful environment and repeated manipulations can reduce the success rate of ET, and recipient selection with an optimal CL size and P4 level can increase the success rate of ET.

## 1. Introduction

Bovine embryo transfer (ET) is one of the most effective ways to produce high-potential calves derived from both in vivo and in vitro embryos, resulting in improved herd genetics. Although the embryo production and transfer processes are cumbersome compared to artificial insemination (AI), they are widely practiced even in repeat breeders with a stable pregnancy rate [1]. However, manipulation of the uterus and damage to the endometrium is inevitable during the ET process, which increases the possibility of an inflammatory response and prostaglandin F2α (PGF2α) secretion [2]. Therefore, the ET operator should select a potential recipient and perform ET in the cranial part of the uterine horn without unnecessary damage [2,3,4].

Using a synchronizing program that can induce ovulation, corpus luteum (CL) development, and consequent hormonal changes in the recipient increases the success rate of ET. Cows with high circulating progesterone (P4) concentrations before super- or single ovulation have a greater percentage of high-grade embryos on the day of recovery than those with low P4 concentrations [5,6]. An increase in P4 after ovulation leads to the expression of endometrial transcriptomes for the successful implantation of bovine embryos, whereas a low concentration of P4 delays uterine receptivity [7]. In addition, elevated P4 after fertilization supports elongation and INF-τ production in the conceptus [8,9,10,11]. Suboptimal P4 concentrations can lead to significant embryo loss within the first three weeks of gestation, during which the maternal endometrium recognizes embryos [12,13].

Small and large CL cells, which are derived from theca and granulosa cells, are the primary sources of P4 secretion in response to luteinizing hormone (LH) and growth hormone (GH) stimulation [14]. A larger CL is believed to produce more P4 in the luteal phase of the estrous cycle [15,16,17], which is supported by blood vessel formation [18]. The area of the CL can be evaluated using ultrasonography, and Doppler ultrasonography is effective for evaluating the blood supply to the functional CL [19,20]. P4 secretion can regulate the early and middle luteal phases by inhibiting PGF2α release, but the regression of CL is initiated by endometrial PGF2α secretion with an inflammatory response [21,22,23]. To increase the conception rate, many studies have provided ways to increase P4 with preovulatory follicles before AI [24,25] and to induce a larger CL, which elevates the P4 level after ovulation either directly or indirectly [15,26,27].

In the present study, we aimed to evaluate the parameters that improve the pregnancy rate of ET, especially with respect to the recipient. After pre-ET treatment using two synchronization programs, we collected data on CL size and P4 concentration the day before ET and evaluated the effects of those measured parameters, along with parity and season, on pregnancy rate.

## 2. Materials and Methods

### 2.1. Animal and Estrus Synchronization

A total of 90 Holstein Friesian cows raised on a free-stall farm at Seoul National University, Pyeongchang, Republic of Korea, were used in the present study. All the medical treatments, as well as the sample collection, were performed at the Farm Animal Medical Teaching Hospital at the request of the University Animal Farm of Seoul National University, where the operator performed ET. From November 2017 to October 2022, 44 of 90 cows were enrolled in synchronization programs more than once in that period; thus, 165 recipient candidates (112 heifers and 53 cows), including mature heifers over 14 months old and healthy cows at least two months after parturition without metritis, mastitis, and lameness, were synchronized for ET, and they were randomly assigned to one of two synchronization protocols (Figure 1) [28,29]. In a conventional protocol (CONV, *n* = 98; 65 heifers and 33 cows), the candidates received the scheduled treatment, by which an intravaginal device with progesterone (CIDR^®^ containing 1.38 g of progesterone, Zoetis, Parsippany, NJ, USA) was inserted on day 0, 2 mg estradiol benzoate (EB; Esron, Samyang, Seoul, Republic of Korea) was injected on day 1, and 500 μg PGF2α (Estrumate^®^, MSD Animal Health, Rahway, NJ, USA) was injected on day 8, along with CIDR removal. Following an injection of 1 mg EB on day 9 and estrus heat detection on day 10, fixed timed embryo transfer (FTET) was performed on day 17. In the J-sync protocol (JSYNC, *n* = 67; 47 heifers and 20 cows), a CIDR insertion was applied with an injection of 2 mg EB on day 0, and it was removed on day 6 with an injection of 500 μg PGF2α. Estrus heat was detected on day 8, 200 μg of gonadotropin-releasing hormone (GnRH; Fertagyl^®^, MSD Animal Health, NJ, USA) was injected on day 9, and FTET was performed on day 16.

### 2.2. Recipient Evaluation

CL sizes were evaluated on the day before ET using B-mode ultrasonography (EASI-SCAN, IMV imaging, Bellshill, UK), and P4 concentrations in plasma collected from coccygeal blood vessels were measured using a fluoroimmunoassay (VetChroma, AniVet, Republic of Korea) in 116 of the 165 recipient candidates. The size of the CL was evaluated in increments of 0.5 cm, and a minimum value was set at 0.5 cm when the response to synchronization was poor and CL development was unclear. The P4 kit was originally developed for canines, but was adjusted by the manufacturer to measure bovine samples. Since concentrations less than 1.0 ng/mL were not displayed, the minimum value was set at 0.5 ng/mL instead of <1.0 ng/mL. The recipients were grouped according to the CL sizes (CL-1, ≤1.5 cm; CL-2, >1.5 cm and ≤2.5 cm; CL-3, >2.5 cm) and the P4 levels (P4-1, ≤2.0 ng/mL; P4-2, >2.0 ng/mL and ≤4.0 ng/mL; P4-3, >4.0 ng/mL), respectively.

### 2.3. Embryo Transfer

Frozen embryos were obtained from the Dairy Cattle Improvement Center of Korea. ET was performed by a trained veterinarian after the recipient’s evaluation. The operator decided whether to perform ET only by rectal palpation of the uterus and ovaries, without evaluation values for recipients. A frozen straw containing an embryo was thawed in a 35 °C water bath for 10 s, and the operator transferred the embryo into the upper one-third of the uterine horn, ipsilateral to CL. ET was performed in 139 (96 heifers and 43 lactating cows; 98 candidates with evaluation records and 41 with only pregnancy diagnosis) out of 165 recipient candidates. Pregnancy was confirmed 40 days after the FTET using B-mode ultrasonography.

### 2.4. Statistical Analysis

A z-test was conducted to evaluate the differences in the ET service rate and pregnancy rate according to synchronization methods, parity of the recipients, and seasons. Differences in P4 levels and CL sizes between selected and unselected recipients and between pregnant and unpregnant recipients were analyzed using the Mann–Whitney rank sum test. The pregnancy rates of the three CL groups and three P4 groups were analyzed using a z-test. Multiple logistic regression analysis was used to predict the pregnancy rate using ET-related variables, including synchronization method, parity of recipient, season, CL size, and plasma P4 level. Statistical analyses were conducted using SigmaPlot 12.5 (Systat Software Inc., San Jose, CA, USA), and significance was set at *p* < 0.05.

## 3. Results

Table 1 shows the results for all recipient candidates who underwent ET synchronization regardless of the CL and P4 evaluation before ET, and their service and pregnancy rates were calculated according to the synchronization methods, parity of recipients, and seasons. The service rates did not differ between synchronization methods (CONV vs. JSYNC), parity (heifers vs. cows), or seasons (from September to February vs. from March to August). There was no statistical difference in pregnancy rates between the two synchronization methods, but in heifers, as well as when ET was performed from September to February, significantly higher pregnancy rates were observed (*p* < 0.05).

The selected recipients showed larger CL sizes and higher P4 concentrations on average, but the difference was not significant (0.065 and 0.349, respectively; Table 2). The average P4 concentration in the pregnant recipients was higher than that in the non-pregnant recipients, but the difference was not statistically significant (*p* = 0.089). The pregnancy rates according to CL size and P4 level are presented in Table 3. The pregnancy rate in the CL-1 group was lower than that in the CL-2 group (*p* = 0.015). There was no statistical difference in pregnancy rates between the groups according to the P4 level (*p* = 0.106), but the pregnancy rate was higher in the P4-2 group than in the other groups.

Among the ET-related factors, the parity of the recipients and the ET seasons significantly influenced the pregnancy rate (*p* = 0.006 and 0.004, respectively; Table 4). Data on the synchronization method and recipient evaluation, including recipients’ CL sizes and plasma P4 levels, was unable to predict the pregnancy rate.

The ultrasound images in each group of CLs on the day before ET are shown in Figure 2. On the day of ET, the operator determined whether the recipient candidates were suitable for various reasons, such as a small CL, poorly developed crown, or too large or soft CL (Figure 2A). The ultrasonograms of the CL in unpregnant and pregnant recipients were also classified according to pregnancy detection and size (Figure 2B). Some candidates with cavitary CL, characterized by a large CL and high P4 level (*n* = 2), were also included in the group of selected recipients (Figure 2C), and none of them were classified as pregnant.

## 4. Discussion

ET is the fastest way to change genetic abilities, allowing for increased productivity in cattle. It is important to increase pregnancy success rates because of the higher expenses and longer time spent on ET compared to AI. Among the factors that affect the success rate of ET, we evaluated the CL sizes and P4 levels, which differed according to recipient selection and pregnancy detection in the present study. The pregnancy rates according to different synchronization methods, parities, and seasons were compared. Difficult cervical passage and longer duration of ET are related to a low pregnancy rate [2,3,4], but variables such as the embryo quality and transfer process were not considered, because a skilled operator transferred only frozen embryos which were validated and provided by the Dairy Cattle Improvement Center.

By inducing cyclicity, several estrus synchronization protocols have been developed that provide additional opportunities for reproductive treatment and pregnancy in dairy and beef cattle. In the present study, two CIDR-based estrous synchronization protocols were applied for the pre-treatment of ET [28,29]. The selection and pregnancy rates of CONV and JSYNC were not statistically different, and both methods effectively induced estrus and CL development; this is represented by the fact that plasma P4 concentrations were higher than 1.0 ng/mL the day before ET in 48/61 recipients of CONV and 40/55 recipients of JSYNC, respectively. The difference between CONV and JSYNC is the period of CIDR insertion, timing of EB administration at the start of the program, and administration of EB or GnRH around estrus. It has been shown that estradiol treatment induces dominant follicle regression and new follicular wave development under conditions in which progesterone concentration is maintained through CIDR insertion [30,31]. In that study, estradiol worked effectively even when administered simultaneously with CIDR insertion or two days later [31]. In several studies, the use of estradiol was preferred for pregnancy rates in buffalo and beef cattle [32,33], but the synchronization protocols of the present study were not intended to show differences between estradiol and GnRH. In JSYNC, GnRH administration was performed after estrus, but the effect varied according to the program [34,35]. In the present study, the CL size and P4 concentration were analyzed only before ET, and the analysis proceeded by assuming that estrus heat and CL development were induced effectively in all candidates after synchronization treatment.

Many previous studies have shown higher pregnancy rates in heifers than in lactating cows [36,37,38]. In one study, the significance varied, but it was proven that repeated synchronization, manipulation, and weight gain decreased fertility in lactating cows [36]. A higher metabolic clearance of P4 and estradiol in lactating cows supports lower pregnancy rates than those in heifers [39,40]. Heat stress also significantly affects reproductive performance in terms of AI, ET, and milk yield after parturition [41,42]. The higher pregnancy rates in autumn and winter than in spring and summer are consistent with previous studies (Table 1), but this can vary depending on local temperature and humidity [36]. Pyeonchang, where the farm is located, has a low average temperature and low humidity every month compared with other areas of South Korea. However, the seasonal effect on the pregnancy rate was significant. Furthermore, vaccination against foot-and-mouth disease, which is administered twice a year in South Korea (in April and November), may have affected the reproductive performance in response to the synchronization program [43], although the plan to synchronize estrus began after vaccination. Kasimanickam et al. [44] reported changes in hormone levels and reproductive performance in response to heat stress, and showed a negative association between prolactin, cortisol, and isoprostane, increasing in response to heat and oxidative stress, with P4.

Recipients were not selected in cases of poor or abnormal CL as diagnosed by rectal palpation, but the statistical significance of the pregnancy rate was not high (Table 2). It was assumed that some of the unselected candidates had a high CL size or P4 concentration due to CL cysts [45], and these results were also reflected in the recipients with CL cysts. However, group comparisons according to CL size showed a significantly higher pregnancy rate in the CL-2 group (*p* < 0.05, Table 3). Palpation of the ovary via rectal examination is an important process for identifying a recipient with an appropriately sized CL, but it is not easy to accurately determine suitability. Therefore, even when selecting a recipient on the day of ET, accurate diagnosis and selection using CL images obtained by ultrasound examination could be used to increase the pregnancy rate. Color Doppler ultrasonography was one of the tools used to effectively evaluate the luteal function around the transfer day [20]. The investigation of blood vessels around CL is closely related to CL development, and is expected to reflect plasma P4 concentration rather than a single measurement of CL size [19]. However, the variation in blood vessel development in the mid-luteal phase and the subjectivity of evaluation are remaining tasks to be solved. A larger CL and higher P4 on the day of ET represent high reproductive performance in cows [3,12,13,36], but the correlation between CL size and P4 was not significant in the present study (r = 0.145, *p* = 0.156). The pregnancy rate with a functional CL depends not only on the size, but also on the shape of the crown and the presence of a cavity [3,46], which also differs according to parity [40]. In addition, it is necessary to analyze the changes in follicular and uterine function in response to circulating P4 to investigate the pregnancy rate with ET [47]. A limitation of this study is that the pregnancy rate was identified only by the CL size and P4 concentration the day before ET, without considering other variables such as uterine environment and milk production. As a result, unlike the analysis results of the recipients’ parity and seasons, neither the CL size nor the P4 concentration one day before ET showed significance in predicting the pregnancy rate after ET. However, we attempted to find a practical index that was easily accessible in the field to select an appropriate recipient for ET. Depending on the different analysis methods, the CL size showed higher significance than the P4 level. However, interestingly, some pregnant recipients had a small CL and high P4 concentrations, but there were also those with P4 concentrations less than 1.0 ng/mL. Regardless of whether a cow is pregnant or not, the P4 level should show a continuous increase from the formation of the CL to the middle of the luteal phase or gestation period [48], but the level at each point can vary between cows. Further studies are required to analyze how the relative changes in CL size and P4 level before and after ET affect pregnancy rates and how to identify them as good ET recipients.

## 5. Conclusions

In conclusion, the parity of the recipient and the ET season had a significant effect on the pregnancy rate with ET, and a CL with a size over 2 cm was associated with a higher pregnancy rate. On the other hand, the effects of P4 the day before ET and of two synchronization methods on pregnancy rate were not statistically significant, but P4 levels tended to be higher in pregnant recipients. Recipient evaluation was performed the day before ET to reduce stress, which could affect the pregnancy rate, and to account for the time spent on P4 analysis. Thus, it could not precisely reflect the state of the recipient on the day of ET, particularly the P4 level. Considering these results, the pregnancy rate of ET can be elevated by the selection of recipients with optimal CL sizes (1.5 < CL ≤ 2.5 cm), and the failure rate of ET could be lowered by excluding recipients with low P4 levels (≤2.0 ng/mL).

## Figures and Tables

**Figure 1 vetsci-10-00262-f001:**
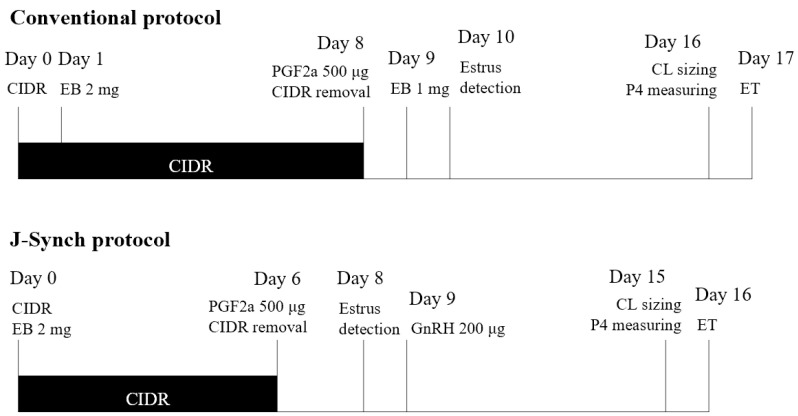
Diagram of two estrus synchronization protocols for embryo transfer (ET) and recipient evaluation with corpus luteum (CL) sizing and progesterone (P4) measuring; CIDR = insertion of an intravaginal device with 1.38 g of progesterone, EB = injection of estradiol benzoate, PGF2α = injection of prostaglandin F2α, GnRH = injection of gonadotropin-releasing hormone.

**Figure 2 vetsci-10-00262-f002:**
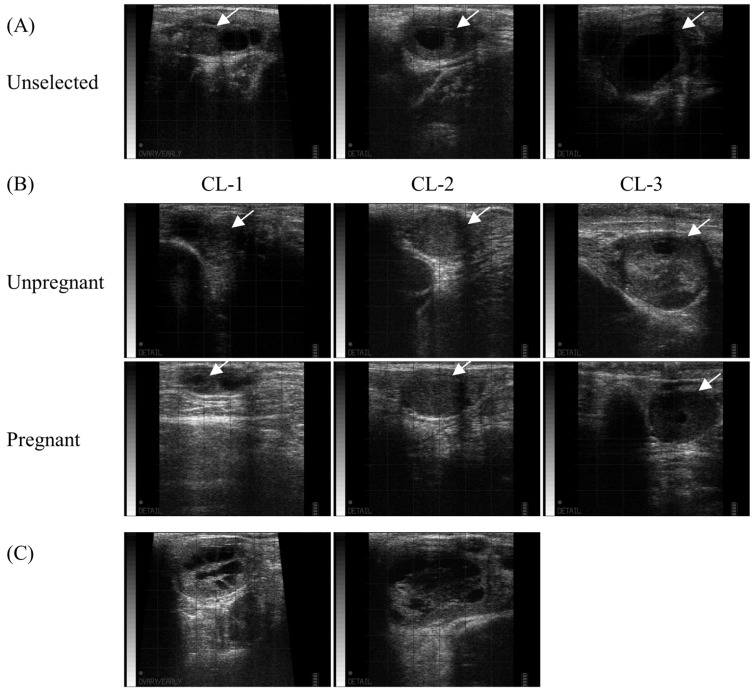
Corpus luteum (CL) of unselected, unpregnant, and pregnant recipients. The ultrasound CL images of were recorded the day before ET, and the operator performed ET without any notification of the evaluation results. (**A**) The candidates with poor ovarian structures palpated by rectal examination were excluded. (**B**) The representative figures of the selected recipients are shown in each category according to the pregnancy and CL size: CL-1 (≤1.5 cm), CL-2 (>1.5 cm and ≤2.5 cm), and CL-3 (>2.5 cm). White arrows indicate the presence of CLs. (**C**) Among the selected recipients, two had cavitary CL with high P4 concentrations, but were not pregnant.

**Table 1 vetsci-10-00262-t001:** Service rate and pregnancy rate according to synchronization protocols, parity, and season in all cows (*n* = 165). Significant differences according to the z-test are expressed with * (*p* < 0.05).

	Candidates (n)	Recipients (n)	SR (%)	Pregnancy (n)	PR (%)
Total	165	139	84.2	65	46.8
CONV	98	82	83.7	40	48.8
JSYNC	67	57	85.1	25	43.9
Heifer	112	96	85.7	53	55.2 *
Cow	53	43	81.1	12	27.9
September~February	87	72	82.8	40	55.6 *
March~August	78	67	85.9	25	37.3

Abbreviations: SR, service rate; PR, pregnancy rate; CONV, conventional protocol; JSYNC, J-sync protocol.

**Table 2 vetsci-10-00262-t002:** CL sizes and P4 levels of cows either unselected or selected by embryo transfer operator (*n* = 116).

	n	CL Size (cm)	*p* Value	P4 Level (ng/mL)	*p* Value
Unselected	18	1.88 ± 0.67	0.065	1.98 ± 2.15	0.349
Selected	98	2.11 ± 0.58		2.54 ± 1.99	
Unpregnant	52	2.10 ± 0.68	0.752	2.25 ± 1.78	0.089
Pregnant	46	2.12 ± 0.47		2.85 ± 2.20	

**Table 3 vetsci-10-00262-t003:** Pregnancy rate (pregnant individuals/total ET recipient) according to the CL size and P4 level in 98 cows. The recipients are grouped into CL-1 (≤1.5 cm), CL-2 (>1.5 cm and ≤2.5 cm), and CL-3 (>2.5 cm) by the CL size, and into P4-1 (≤2.0 ng/mL), P4-2 (>2.0 ng/mL and ≤4.0 ng/mL), and P4-3 (>4.0 ng/mL) by the P4 level. Different superscripts within a row and column indicate significant differences between groups, as revealed by z-test.

	CL-1	CL-2	CL-3	Total
P4-1	26.7% (4/15)	46.2% (12/26)	16.7% (1/6)	36.2% (17/47) ^a^
P4-2	25.0% (1/4)	71.4% (15/21)	40.0% (2/5)	60.0% (18/30) ^a^
P4-3	33.3% (1/3)	60.0% (9/15)	33.3% (1/3)	52.4% (11/21) ^a^
Total	27.3% (6/22) ^a^	58.1% (36/62) ^b^	28.6% (4/14) ^ab^	46.9% (46/98)

**Table 4 vetsci-10-00262-t004:** Multiple logistic regression analysis of embryo transfer variables with respect to pregnancy rate.

Variables	OR	±95% CI	*p* Value
Synchronization	0.648	0.234–1.796	0.405
Parity	0.246	0.081–0.752	0.014
Season	5.668	2.002–16.045	0.001
CL size	1.145	0.473–2.776	0.764
P4 level	1.284	0.695–2.374	0.424

Abbreviations: OR (odds ratio); CI (confidence interval).

## Data Availability

Not applicable.

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
