# Peer review of "Evaluation of Corpus Luteum and Plasma Progesterone the Day before Embryo Transfer as an Index for Recipient Selection in Dairy Cows"

_vetsci, 2023, doi:10.3390/vetsci10040262_

Round 1

Reviewer 1 Report

This manuscript aims to determine some factors that influence the results of embryo transfer: CL size, plasma P4 concentration, synchronization protocol, parity and season. Although it is interesting to know how to improve fertility of the recipients, the factors studied have already been evaluated in previous research. Nowadays, regarding ultrasonography, published studies are focused on the use of Doppler mode to assess CL blood flow and, in some cases, the cavity of the CL. Concerning parity and season, these are factors that have been intensively studied regarding their effect on fertility in general and recipients in particular, as well as synchronization protocols. Therefore, this manuscript lacks novelty and its contribution to the field is rather scarce. Maybe it can be considered for publication, after a thorough review, as a short communication.

Abstract

L24: 2.0 – 4.0 ng/mL

Introduction

L59 – 60: pregnancy rate?

Materials & Methods

L65 – 80: please, indicate the number of cows and heifers allocated in each treatment group. Also indicate the criteria for allocation.

L86 – 97: make clear the number of animals. 90 cows (L66), 165 recipients, 116 evaluations? Why do the numbers not match? Also explain the selection criteria for recipients. Explain the characteristics of the recipients: BCS, milk production...

L98 – 105: describe the process in detail: technique, embryos used, operator...

L106 – 113: to stablish the influence of a factor when there are others that may also influence, the best approach is to perform multiple analysis in which all factors are included. In this particular case, a multivariable logistic regression may be interesting, so the confusing effect of parity and season may be corrected, and the final result may vary. On the other hand, if the operator that performs the evaluation is not always the same, there might be an interaction between service rate and fertility, as a more demanding operator would exclude more animals and their fertility would be higher.

Results

Figure 2: it is not clear that CL type C are cystic CL, they seem more like CL with a cavity, and the presence of a cavity does not imply lower P4 production.

Discussion

L163 – 166: does the fact that they are imported embryos prevent from having data about the transfer process? Those data are important, as they may significantly vary fertility.

L209 – 213: as nowadays the vascularization of the CL can be assessed and it is related to plasma P4 levels, Doppler mode should be mentioned in this section. Moreover, in the recent years, the studies about the effect of CL on fertility in recipients are based on Doppler mode.

Conclusion

Only CL size and P4 levels are mentioned. Conclusions about synchronization method, parity and season are missing. In the abstract, all factors are mentioned, but not in the text.

Author Response

Reviewer #1

This manuscript aims to determine some factors that influence the results of embryo transfer: CL size, plasma P4 concentration, synchronization protocol, parity and season. Although it is interesting to know how to improve fertility of the recipients, the factors studied have already been evaluated in previous research. Nowadays, regarding ultrasonography, published studies are focused on the use of Doppler mode to assess CL blood flow and, in some cases, the cavity of the CL. Concerning parity and season, these are factors that have been intensively studied regarding their effect on fertility in general and recipients in particular, as well as synchronization protocols. Therefore, this manuscript lacks novelty and its contribution to the field is rather scarce. Maybe it can be considered for publication, after a thorough review, as a short communication.

: Thank you for your valuable comments regarding our research paper. Many studies have evaluated the factors related to the pregnancy rate of ET, and we agree that those factors include blood flow and cavity size of the CL. However, in the present study, we evaluated the factors one day before ET, and it was focused on the size of the CL and especially the concentration of P4. Although we have not presented the results of the repeated evaluation for an extended period, we believe that our research can provide a meaningful basis for selecting recipients for ET. We would be happy to make further corrections if necessary and look forward to hearing from you. As you suggested, we've revised the conclusion to demonstrate the novelty and value of the paper.

Abstract

L24: 2.0 – 4.0 ng/mL

: As you suggested, “between 2.0 4.0” was revised to “between 2.0 and 4.0” (L24).

Introduction

L59 – 60: pregnancy rate?

: Thank you for your comment. As you suggested, we’ve evaluated the pregnancy rate rather than conception rate since pregnancy was confirmed after 40 days of ET. We’ve replaced “conception” to “pregnancy” throughout the manuscript (L62, 118, 122 and 195).

Materials & Methods

L65 – 80: please, indicate the number of cows and heifers allocated in each treatment group. Also indicate the criteria for allocation.

In each synchronization protocol, 65 heifers and 33 cows in CONV (L78), and 47 heifers and 20 cows in JSYNC (L85) were randomly allocated to each group (L77). We’ve detailed the number of animals in the manuscript.

L86 – 97: make clear the number of animals. 90 cows (L66), 165 recipients, 116 evaluations? Why do the numbers not match? Also explain the selection criteria for recipients. Explain the characteristics of the recipients: BCS, milk production...

: We would like to apologize for making you confused and have revised the manuscript. 44 of 90 cows were used as a recipient more than once, so the number of recipient candidates is larger than animals (L73 – 85). Among the candidates, evaluation data of 116 were used for the results of Table 2 and 3. The remaining 49 candidates were used for estrus synchronization, and 41 among them were used as ET recipients, but they only had the results of pregnancy diagnosis because of technical issues. At the time of ET treatment, mature heifers over 14 months old and healthy cows at least two months after parturition without metritis, mastitis, and lameness were treated for estrus synchronization (L75 – 76), and the recipients were selected after rectal palpation by the operator on the day of ET.

L98 – 105: describe the process in detail: technique, embryos used, operator...

: We also agree that the ET process needs to be described. The explanation was added to the manuscript; “A frozen straw containing one blastocyst was thawed in a 35°C water bath for 10 sec, and the operator transferred the embryo into upper one-third of the uterine horn ipsilateral to CL.” (L111 – 113)

L106 – 113: to stablish the influence of a factor when there are others that may also influence, the best approach is to perform multiple analysis in which all factors are included. In this particular case, a multivariable logistic regression may be interesting, so the confusing effect of parity and season may be corrected, and the final result may vary. On the other hand, if the operator that performs the evaluation is not always the same, there might be an interaction between service rate and fertility, as a more demanding operator would exclude more animals and their fertility would be higher.

: Thank you for your valuable comments. We’ve conducted binary logistic regression analysis, and among the variables, P values were high in CL size and P4 level. Thus, in order to compare the pregnancy rate among the groups according to CL and P4, a z-test was conducted. We determined to add table 4 to clarify the influence of variables. However, one operator performed all ET process, so that was not considered. (L158, Table 4 and L255 – 259)

Results

Figure 2: it is not clear that CL type C are cystic CL, they seem more like CL with a cavity, and the presence of a cavity does not imply lower P4 production.

: We would like to apologize for making you confused and have revised the manuscript and figure legend (L172 and L182). Figure 2C shows representative ultrasound images with a sufficiently large CL and high P4, but not pregnant. CL with a cavity was not considered as a variable, so it was not mentioned in the other part of results.

Discussion

L163 – 166: does the fact that they are imported embryos prevent from having data about the transfer process? Those data are important, as they may significantly vary fertility.

: We agree that there should be explanation embryos used in this study. However, they were imported and provided through a verified route by a Dairy Cattle Improvement Center, and sample evaluation of embryos were conducted by the institution, so it was not conducted in this paper (L192).

L209 – 213: as nowadays the vascularization of the CL can be assessed and it is related to plasma P4 levels, Doppler mode should be mentioned in this section. Moreover, in the recent years, the studies about the effect of CL on fertility in recipients are based on Doppler mode.

: We also agree that color Doppler US is an effective tool for functional evaluation of CL and further study using Doppler is required in dairy cow’s ET. We’ve added some discussion to L240 – 245.

Conclusion

Only CL size and P4 levels are mentioned. Conclusions about synchronization method, parity and season are missing. In the abstract, all factors are mentioned, but not in the text.

: We noticed that we were missing the conclusion about synchronization method, parity and season, and we revised the conclusion part (L268 – 274).

Reviewer 2 Report

The authors investigated the use CL size and plasma progesterone as indices to optimize the success of embryo transfer using 2 different estrous synchronization protocols during different seasons of the year. Authors determined that the season ET was implemented and CL size influenced the successful outcomes of ET.  The abstract and introduction sections require a thorough editing for English grammar due to poor sentence structure and punctuation issues.  The rest of the text did not have as many problems. Some other concerns are below:

1.     Adjust table to column titles, currently too crammed

2.     Line 29-3: restructure sentence, doesn’t make sense in current format.

3.     Line 41: ‘greater percentage of high-grade day 7 embryos’…after AI?

4.     Line 42: endometrial transcriptomes that enhance the success implantation?

5.     Line 66: author stated only 90 total cows were used but the numbers in ‘embryo transfer’ section 2.3 have different values and in table 1. Correction or clarification is required.

6.     Line 41: significant p value needs to be included after the word ‘significant’

Author Response

Reviewer #2

The authors investigated the use CL size and plasma progesterone as indices to optimize the success of embryo transfer using 2 different estrous synchronization protocols during different seasons of the year. Authors determined that the season ET was implemented and CL size influenced the successful outcomes of ET.  The abstract and introduction sections require a thorough editing for English grammar due to poor sentence structure and punctuation issues.  The rest of the text did not have as many problems. Some other concerns are below:

  1. Adjust table to column titles, currently too crammed
    ; Thank you for review our manuscript and kind advice. We agree that Table 1 is hard to read, so we added some abbreviations and revised (L139, Table 1). If there is another table to be revised, please let us know.

  1. Line 29-3: restructure sentence, doesn’t make sense in current format.

The sentence has been restructured (L29 – 31).

  1. Line 41: ‘greater percentage of high-grade day 7 embryos’…after AI?

We would like to apologize for making you confused and the sentence have been revised (L42 – 44).

  1. Line 42: endometrial transcriptomes that enhance the success implantation?

Thank you for your kind advice. “For bovine embryo implantation” has been replaced with “for the successful implantation of bovine embryo” (L44 – 46).

  1. Line 66: author stated only 90 total cows were used but the numbers in ‘embryo transfer’ section 2.3 have different values and in table 1. Correction or clarification is required.

I’m sorry to make you confused. 44 of 90 cows were used as a recipient more than once, so the number of recipient candidates is larger than animals. The explanation was added to M&M 2.1 part (L73 – 85).

  1. Line 41: significant p value needs to be included after the word ‘significant’

I`m sorry but we couldn’t find the word “significant” on L41 and other parts needs p value. It would be appreciable if you let us know other parts that p value should be added.

Reviewer 3 Report

The study of Choi et al. attempts to evaluate risk factors for a successful ET in dairy cows. Unfortunately, the study design has so many limitations that in my opinion the findings of the present study offer no important nor solid scientific knowledge. Additionally, there is no novelty in the study and it has no clear aim, for instance the authors test two synchronization protocols with absolutely no justification or physiological background of these protocols. There has been no randomization at all in the assignment to treatments. On the contrary, there has been an arbitrary selection of recipient cows by an “operator” (obviously the authors just measured cows for ET under commercial and not research conditions). There has also been an arbitrary classification of CL size and P4 values without any biological background. The statistical analysis is very poor and improper, as z-test is an equivalent to t-test and is used to determine whether two population means are different when the variances are known and the sample size is large. I do not understand how it was used to compare proportions. Even so, a study design as described here would demand a multivariate analysis in order to check for possible interactions and confounding factors.   

Author Response

Reviewer #3

The study of Choi et al. attempts to evaluate risk factors for a successful ET in dairy cows. Unfortunately, the study design has so many limitations that in my opinion the findings of the present study offer no important nor solid scientific knowledge. Additionally, there is no novelty in the study and it has no clear aim, for instance the authors test two synchronization protocols with absolutely no justification or physiological background of these protocols. There has been no randomization at all in the assignment to treatments. On the contrary, there has been an arbitrary selection of recipient cows by an “operator” (obviously the authors just measured cows for ET under commercial and not research conditions). There has also been an arbitrary classification of CL size and P4 values without any biological background. The statistical analysis is very poor and improper, as z-test is an equivalent to t-test and is used to determine whether two population means are different when the variances are known and the sample size is large. I do not understand how it was used to compare proportions. Even so, a study design as described here would demand a multivariate analysis in order to check for possible interactions and confounding factors. 

; Thank you for your very careful review of our manuscript. In this study, we aimed to clarify the effect of synchronization method, season, parity, corpus luteum (CL) size, and progesterone (P4) levels on the pregnancy rate of ET. In particular, it was to analyze whether selecting a recipient with a high probability was possible using the evaluation data the day before ET. despite the limitations of the study design, the classification of CL and P4 values was set to obtain one group which yields the highest pregnancy rate. Unfortunately, significances were found only in the results of parity and season (multiple logistic regression), and a significance according to CL size was obtained through z-test. However, through the pregnancy rate and significance of each group, we believe that a basis for using the evaluation data the day before ET to select a recipient was provided. By adding the results of multiple logistic regression, we've revised the manuscript thoroughly. It would be appreciable if you let us know the way to improve the manuscript.

Reviewer 4 Report

This manuscript describes a five year study measuring progesterone levels and Corpus luteal sizes in relation to resulting pregnancy at Day 40, after two synchronisation protocols and ET transfer to heifers and lactating parous cows. As has been demonstrated before, heifers are found to be superior recipients while time of year has a significant though unexplained effect too. The data is clearly presented and the paper is well-written. My concern is that the statistical analysis mainly for Table 3 is inadequate, as the significant effects of cow/heifer and season as well as the interactive effect of progesterone versus corpus luteal size are not addressed in the simple z-test analyses. Probably a mixed linear model would be more appropriate. For example, any biased distribution in parous cows in the CL1 and Cl2 populations might account for the significant differences in the pregnancy rates. Furthermore, a more sophisticated statistical model might tease out the best combination of CL size and progesterone.

Author Response

Reviewer #4

This manuscript describes a five year study measuring progesterone levels and Corpus luteal sizes in relation to resulting pregnancy at Day 40, after two synchronisation protocols and ET transfer to heifers and lactating parous cows. As has been demonstrated before, heifers are found to be superior recipients while time of year has a significant though unexplained effect too. The data is clearly presented and the paper is well-written. My concern is that the statistical analysis mainly for Table 3 is inadequate, as the significant effects of cow/heifer and season as well as the interactive effect of progesterone versus corpus luteal size are not addressed in the simple z-test analyses. Probably a mixed linear model would be more appropriate. For example, any biased distribution in parous cows in the CL1 and Cl2 populations might account for the significant differences in the pregnancy rates. Furthermore, a more sophisticated statistical model might tease out the best combination of CL size and progesterone.

; Thank you for review our manuscript and kind advice. We’ve added table 4, the results of multiple logistic regression, to show the effect of all ET-related variables (L158, Table 4 and L255 – 259). Unfortunately, significances were found only in parity and season (multiple logistic regression), and a significance according to CL size was obtained through z-test.

Round 2

Reviewer 3 Report

The authors have only added a logistic regression in their analysis, which also did not include two way interactions and as a result no new finding is present in the revised form of the manuscript. Additionally, the major flaw of the study design, which is randomization in treatments and background as to why these treatments were chosen remain unsolved.

Author Response

Thank you for your kind suggestion. I would like to clarify that in our study, we used chi-square analysis and logistic regression analysis with pregnancy status as the dependent variable, rather than pregnancy rate. We believe that this approach is appropriate for analyzing our results. We appreciate your recommendation to explore variables and interactions related to ET pregnancy rates, and we will always make sure to carefully consider and select an appropriate analytical method.